# Speech Reconstruction from Silent Lip and Tongue Articulation by Diffusion Models and Text-Guided Pseudo Target Generation

## ABSTRACT

This paper studies the task of speech reconstruction from ultrasound tongue images and optical lip videos recorded in a silent speaking mode, where people only activate their intra-oral and extra-oral articulators without producing real speech. This task falls under the umbrella of articulatory-to-acoustic (A2A) conversion and may also be referred to as a silent speech interface. To overcome the domain discrepancy between silent and standard vocalized articulation, we introduce a novel pseudo target generation strategy. It integrates the text modality to align with articulatory movements, thereby guiding the generation of pseudo acoustic features for supervised training on speech reconstruction from silent articulation. Furthermore, we propose to employ a denoising diffusion probabilistic model as the fundamental architecture for the A2A conversion task and train the model using a combined training approach with the generated pseudo acoustic features. Experiments show that our proposed method significantly improves the intelligibility and naturalness of the reconstructed speech in the silent speaking mode compared to all baseline methods. Specifically, the word error rate of the reconstructed speech decreases by approximately 5% when measured using an automatic speech recognition engine for intelligibility assessment, and the subjective mean opinion score for naturalness improves by 0.14. Moreover, analytical experiments reveal that the proposed pseudo target generation strategy can generate pseudo acoustic features that synchronize better with articulatory movements than previous strategies. Samples are available at our project page[1].

## CCS CONCEPTS

• **Information systems** → **Multimedia content creation**; • **Computing methodologies** → **Natural language generation**.

## KEYWORDS

articulatory-to-acoustic conversion, silent speech interface, diffusion probablistic model, pseudo target

## 1 INTRODUCTION

The human speech production process intricately involves the coordination of various vocal organs, particularly the collaboration between the lips and tongue, which manipulate the shape of the

---

[1]Samples are available at https://anonymous.4open.science/w/Diff-A2A-F494/.

*ACM MM, 2024, Melbourne, Australia*
© 2024 Copyright held by the owner/author(s). Publication rights licensed to ACM.
ACM ISBN 978-x-xxxx-xxxx-x/YY/MM
https://doi.org/10.1145/nnnnnnn.nnnnnnn

vocal tract to produce different phonemes. Therefore, articulatory and acoustic features are intrinsically linked [26]. Motivated by the above theory, this paper studies the articulatory-to-acoustic (A2A) conversion task, focusing on reconstructing speech from lip videos and ultrasound tongue images [3, 18]. This task also falls under the umbrella of silent speech interfaces (SSIs), which rely on non-acoustic signals generated during the speech production process to enable communication in scenarios where regular verbal communication is impossible [5, 7, 38]. The exploration of A2A conversion not only deepens the understanding of speech production mechanisms but also provides diverse practical applications, such as facilitating speech communication for patients with dysphonia and assisting communication when speech is not available or desirable.

Over the past few years, there has been a great deal of work studying speech reconstruction from either the tongue[4, 8, 21, 43], lip[9–11, 19, 20, 44], or a combination of both [15, 16, 49]. These studies mainly look at a standard vocalized speaking mode, where speakers' larynx and lungs function normally, and they naturally receive auditory feedback while speaking. However, people adopt different speaking modes in different scenarios. Under some circumstances where silence is required, or for some laryngectomy patients, speakers tend to utilize a silent speaking mode instead of the standard vocalized mode. In this mode, speakers only activate their oral and nasal articulators but suppress their laryngeal activity, and consequently, no speech is produced as output. Reconstructing speech from silent articulation faces the following two challenges. First, models trained on the vocalized data cannot be directly applied to the silent mode due to the domain discrepancy between vocalized and silent articulation, including incomplete, reduced, and prolonged articulator movements in the silent speaking mode [6, 36, 41, 42, 49]. Second, since no speech signals are produced in the silent speaking mode, traditional supervised training paradigms cannot be directly applied to training models with silent articulation as input. A previous study has proposed to employ pseudo target generation, accompanied by domain adversarial training and iterative training strategy [50] to address these challenges, showcasing certain improvements of speech reconstruction in the silent speaking mode.

Nevertheless, the discrepancy between vocalized and silent articulation can lead to low-quality pseudo targets generated by the previous strategy [50], thereby impacting the overall performance of reconstructing speech from silent articulation. To overcome this challenge, this paper introduces a novel pseudo target generation strategy, named the dubbing strategy. This strategy integrates a new text modality to describe the linguistic content of the silent articulation, without resorting to cumbersome iterative methods or complex adversarial training strategies to learn from corresponding vocalized articulation, as utilized in previous work [50]. Specifically, by learning the alignment between text and articulation, the dubbing

strategy generates pseudo acoustic features synchronized better with the given articulatory movement than previous method while maintaining content consistency with the provided text, thereby improving the overall task performance. Notably, as text information is usually unfeasible in practical applications, the generated pseudo acoustic features serve solely as supervision targets for training A2A conversion models in the silent speaking mode.

Furthermore, we propose an A2A conversion architecture based on a denoising diffusion probabilistic model (DDPM) [12], which is conditioned on lip and tongue articulatory representations. DDPMs, abbreviated as diffusion models, have obtained state-of-the-art performance across various speech generation tasks, including neural vocoder [1, 22], speech enhancement [30, 47], and text-to-speech (TTS) synthesis [14, 17, 24, 25]. In line with these methods, we construct a diffusion-based A2A conversion architecture. Specifically, the proposed architecture involves an articulation encoder for transforming lip videos and ultrasound tongue images into hidden articulatory representations and a diffusion-based spectrogram denoiser to synthesize acoustic features from random noise conditioned on these hidden representations step-by-step. Our proposed diffusion-based architecture demonstrates the ability to generate less over-smoothing and more diverse acoustic features than previous non-probabilistic generative models. Moreover, training the proposed diffusion-based A2A conversion model with a combination of pseudo acoustic features generated by different pseudo target generation strategies can further improve the naturalness and intelligibility of the speech reconstructed from the silent lip and tongue articulation, as proven by the experimental results.

The main contributions of this paper are summarized as follows:

(1) To overcome the domain discrepancy between vocalized and silent articulation, we introduce a novel pseudo target generation strategy, integrating the text modality to guide the generation of pseudo acoustic features for supervised training on speech reconstruction from silent articulation.

(2) We propose a diffusion-based A2A conversion model as the fundamental architecture for reconstructing speech from lip videos and ultrasound tongue images. Besides, a combined training approach is proposed to further improve the naturalness and intelligibility of the reconstructed speech in the silent mode.

(3) Experimental results demonstrate that our proposed method enhances the naturalness and intelligibility of the speech reconstructed from lip videos and ultrasound tongue images in the silent speaking mode. In addition, analytical experiments reveal that the proposed dubbing strategy can generate pseudo acoustic features that synchronize better with articulatory movements than the previous method [50].

## 2 RELATED WORK

### 2.1 Diffusion-based TTS models

Diffusion models have achieved state-of-the-art performance across various speech generation tasks, particularly in TTS [14, 17, 24, 25], where they usually serve as the decoder to transform text embeddings into acoustic features. These diffusion models typically comprise a forward diffusion process and a reverse denoising process. The diffusion process is defined by a fixed $T$-step Markov chain

from initial data $\mathbf{x}_0$ to the latent variable $\mathbf{x}_T \sim \mathcal{N}(0, \mathbf{I})$ as follows:

$$
\begin{aligned}
q(\mathbf{x}_1, \cdots, \mathbf{x}_T | \mathbf{x}_0) &= \prod_{t=1}^{T} q(\mathbf{x}_t | \mathbf{x}_{t-1}) \\
&= \prod_{t=1}^{T} \mathcal{N}(\mathbf{x}_t; \sqrt{1 - \beta_t} \mathbf{x}_{t-1}, \beta_t \mathbf{I}),
\end{aligned}
\tag{1}
$$

which gradually converts the data $\mathbf{x}_0$ to whitened latent $\mathbf{x}_T$ by adding small random noise according to a predefined noise schedule $\{\beta_t\}_{t=1}^{T}$. The reverse denoising process is a Markov chain from $\mathbf{x}_T$ to $\mathbf{x}_0$, parameterized by shared $\theta$, which aims to recover samples from Gaussian noises:

$$
\begin{aligned}
p_\theta(\mathbf{x}_0, \cdots, \mathbf{x}_{T-1} | \mathbf{x}_T) &= \prod_{t=1}^{T} p_\theta(\mathbf{x}_{t-1} | \mathbf{x}_t) \\
&= \prod_{t=1}^{T} \mathcal{N}(\mathbf{x}_{t-1}; \mu_\theta(\mathbf{x}_t, t), \sigma_t^2 \mathbf{I}),
\end{aligned}
\tag{2}
$$

where $\mu_\theta(\mathbf{x}_t, t)$ and $\sigma_t^2$ are the mean and variance of the added Gaussian noise at $t$-th step, respectively.

Current diffusion-based TTS models can be classified into two main categories: gradient-based models and generator-based models [14]. Gradient-based TTS models [17, 22, 24] parameterize the denoising model $\theta$ by predicting Gaussian noises $\epsilon$ in the diffusion process with a neural network $\epsilon_\theta$. Therefore, the training loss function is defined as the mean squared error in the $\epsilon$ space. However, these gradient-based TTS models usually require hundreds of thousands of denoising steps to guarantee high sample quality, leading to substantial computational costs. Different from gradient-based TTS models, generator-based TTS models [14, 25] directly predict clean data $\mathbf{x}_0$ with a neural network $f_\theta$ and then add back perturbation using the posterior distribution $q(\mathbf{x}_{t-1} | \mathbf{x}_t, \mathbf{x}_0)$. In this case, the training loss function is defined as the mean squared error in the data $\mathbf{x}_0$ space. These generator-based TTS models have the advantage of expediting sampling from a complex distribution while retaining satisfactory TTS performance. In this paper, we construct our proposed A2A conversion architecture based on a generator-based diffusion model.

### 2.2 Speech Reconstruction from Lip and Tongue Articulation

TaLNet [49] currently stands as the state-of-the-art model for speech reconstruction from ultrasound tongue images and lip videos in the vocalized speaking mode on Tongue and Lip (TaL) dataset [37] with an encoder-decoder architecture. The encoder of TaLNet first encodes the input tongue images and lip videos into articulatory hidden representations, which are then decoded into acoustic features through a decoder. The produced acoustic features are ultimately fed into a well-trained neural vocoder to synthesize the final speech waveforms. The decoder of TaLNet is migrated from a Tacotron2-based TTS acoustic model [39]. To train TaLNet, a multi-speaker Tacotron2 model is first built on a multi-speaker TTS corpus. Then, its decoder is transferred as a TaLNet decoder and jointly trained with the encoder of TaLNet.

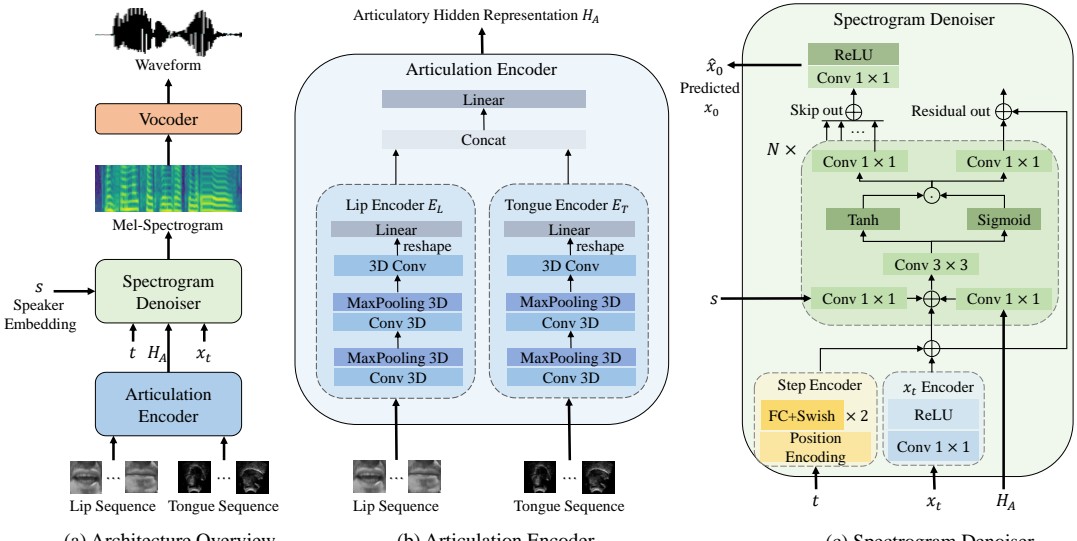

**Figure 1: Proposed A2A Conversion Architecture**

Since TaLNet [49] focuses on speech reconstruction from vocalized articulation, it suffers from significant performance degradation on silent articulation. Zheng et al. [50] have proposed further enhancements for TaLNet [49] and achieved state-of-the-art results of speech reconstruction in the silent speaking mode on the TaL dataset [36]. Their method utilizes dynamic time warping (DTW) to align the articulatory representations outputted by the TaLNet encoder from corresponding vocalized and silent articulation whose linguistic contents are the same. The acoustic features of the vocalized utterance are then aligned based on the alignment path to obtain pseudo acoustic features and facilitate supervised training on silent articulation. Additionally, their method incorporates a domain discriminator to encourage the encoder to learn articulation representations robust in both vocalized and silent domains. Finally, iteratively conducting pseudo target generation and domain adversarial training are suggested to generate high-quality pseudo acoustic targets. Although their method has improved the model's performance on silent articulation, it relies on complex adversarial training strategies and intricate iterative steps, indicating potential areas for further optimization. In particular, the DTW-based pseudo target generation strategy depends on the alignment between silent and vocalized articulation but overlooks their discrepancy. To address this issue, we propose a novel text-guided pseudo target generation strategy, resulting in pseudo acoustic features well-synchronized with the silent articulatory movements.

## 3 PROPOSED METHOD

In this paper, we propose a new A2A conversion architecture based on a diffusion model. The architecture is detailed in Fig. 1. To overcome the discrepancy between vocalized and silent articulation, we introduce a novel pseudo target generation strategy, named dubbing strategy, to synthesize synchronized acoustic features for silent articulation under the guidance of text. Further details are depicted in Fig. 2. The proposed diffusion-based model is then trained with the pseudo acoustic features generated by the dubbing

strategy using a combined training approach. We will introduce each component in this section.

### 3.1 Diffusion-based A2A Conversion Model

The proposed A2A conversion model has an encoder-decoder framework comprising an articulation encoder and a spectrogram denoiser, as shown in Fig. 1(a). Initially, the articulation encoder converts input lip videos and ultrasound tongue images into articulatory hidden representations. Subsequently, the spectrogram denoiser generates predicted acoustic features conditioned on the articulatory hidden representations. Finally, the generated acoustic features are converted into speech waveforms through a vocoder.

*3.1.1 Articulation Encoder.* The structure of the encoder mirrors that of the TaLNet [49] encoder, which includes two identical parallel sub-encoders designed for processing ultrasound tongue images $I_{ton} = [I_{ton,1}, \cdots, I_{ton,F}]$ and optical lip videos $I_{lip} = [I_{lip,1}, \cdots, I_{lip,F}]$, where $F$ represents the length of input articulation frames. Each sub-encoder consists of several stacked 3D convolutional and MaxPooling layers, as illustrated in Fig. 1(b). For both tongue and lip frames, pixel-wise mean and standard deviation are computed for each speaker, repeated, and then appended as extra channels to the ultrasound and lip sequences. Therefore, the resulting input is of dimension $3 \times F \times H \times W$, where $H$ and $W$ denote the height and width of the lip and tongue images. Within the sub-encoder processing, the spatial dimensions $H$ and $W$ are reduced while the time dimension $F$ is preserved. The final convolutional layer outputs are flattened along the time axis and pass through a linear layer to produce a single vector for each frame. Lastly, the vectors from each sub-encoder are fused and passed through a fully connected layer to yield the final hidden representations $\{H_A\}_{i=1}^{F} \in \mathbb{R}^{F \times D}$, where $D$ is the feature dimension.

*3.1.2 Spectrogram Denoiser.* The spectrogram denoiser adopts a similar architecture to the acoustic models in diffusion-based TTS models [14, 25], as illustrated in Fig. 1(c). It employs a noncausal

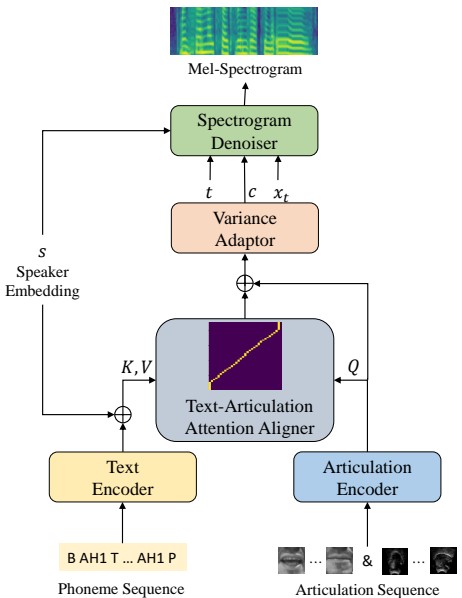

**Figure 2: Pseudo Target Generation Module**

WaveNet architecture [33], consisting of a $1 \times 1$ convolutional layer and $N$ convolution blocks with residual connections and projecting the input articulatory representations with $D$ channels. All residual blocks have a CNN-based speaker embedding transforming block.

The spectrogram denoiser, parameterized in a generator-based manner, iteratively refines the articulatory representations into acoustic features. Specifically, instead of directly modelling $p_\theta(\mathbf{x}_{t-1}|\mathbf{x}_t)$ by predicting $\mathbf{x}_{t-1}$ from $\mathbf{x}_t$, the denoising function is modeled as $p_\theta(\mathbf{x}_{t-1}|\mathbf{x}_t) = q(\mathbf{x}_{t-1}|\mathbf{x}_t, \mathbf{x}_0)$, where $\mathbf{x}_0$ is predicted from diffused sample $\mathbf{x}_t$ by the denoising function $f_\theta$ parameterized with $\theta$. During training, a random step $t$ is first sampled uniformly from $[0, \cdots, T]$, and $\mathbf{x}_t$ are sampled according the following equation

$$q(\mathbf{x}_t|\mathbf{x}_0) = \mathcal{N}(\mathbf{x}_t; \alpha_t \mathbf{x}_0, \sqrt{1 - \alpha_t^2}\mathbf{I})), \quad (3)$$

where $\alpha_t = \prod_{t=1}^{t} \sqrt{1 - \beta_i}$. Next, the sampled $t$ and $\mathbf{x}_t$ are input to the spectrogram denoiser together with the speaker embedding $s$ and articulatory representations $\mathbf{H}_A$ to predict the initial data point $\hat{x}_0 = f_\theta(\mathbf{x}_t|t, s, \mathbf{H}_A)$. The spectrogram denoiser with parameters $\theta$ is trained with the following loss function

$$\mathcal{L}_\theta = ||f_\theta(\alpha_t \mathbf{x}_0 + \sqrt{1 - \alpha_t^2}\epsilon, t, s, \mathbf{H}_A) - \mathbf{x}_0||_2^2, \epsilon \sim \mathcal{N}(0, I). \quad (4)$$

During inference, the spectrogram denoiser $f_\theta(\mathbf{x}_t|t, s, \mathbf{H}_A)$ first predicts $\hat{x}_0$, and then $\mathbf{x}_{t-1}$ is sampled using the posterior distribution $q(\mathbf{x}_{t-1}|\mathbf{x}_t, \mathbf{x}_0)$. As $t$ gradually decreases from $T$ to 1, the final predicted $\mathbf{x}_0$ is obtained.

## 3.2 The Dubbing Strategy for Pseudo Target Generation

We introduce a novel pseudo target generation strategy, named the dubbing strategy, to apply the above A2A conversion architecture for supervised training on silent articulation due to the absence of audible speech in the silent mode. The previous DTW-based pseudo target generation strategy suffers from the discrepancy between

vocalized and silent articulation and thus affects the overall performance of the system. To address this, we incorporate text information to produce pseudo acoustic features. Specifically, the pseudo acoustic features generated for silent articulation maintain linguistic content consistency with the provided phoneme sequences and synchronize their duration with lip and tongue movements.

*3.2.1 Overall Pipeline.* Our proposed pseudo target generation module, depicted in Figure 2, follows a pipeline similar to automatic video dubbing tasks [13, 28, 29]. Taking phoneme sequences and articulatory movements (lip videos and ultrasound tongue images) as input, it align their representations using the text-articulation attention aligner, resulting in expanded phoneme representations whose length equals to that of the input articulatory movements. Next, the expanded phoneme representations are processed by a variance adaptor and then used as the condition $C$ for the spectrogram denoiser to generate acoustic features. The structure of the articulation encoder and spectrogram denoiser in the proposed module is the same as those in Fig. 1. The text encoder is identical to that in FastSpeech2 [35], while the variance adaptor comprises a pitch predictor from FastSpeech2 [35]. The text-articulation aligner is the most vital part in this module, as it establishes correspondence between text information and articulatory movements, controlling the quality of the produced acoustic features.

*3.2.2 Text-Articulation Aligner.* The text-articulation aligner comprises an scaled dot-product attention module as in automatic video dubbing tasks [13, 29]. In the attention module, articulatory hidden representation $\mathbf{H}_A$ serves as the query, and the phoneme hidden representation $\mathbf{H}_P$ output by the text encoder is used as both the key and the value:

$$\begin{aligned} Attention(Q, K, V) &= Attention(\mathbf{H}_A, \mathbf{H}_P, \mathbf{H}_P) \\ &= Softmax\left(\frac{\mathbf{H}_A \mathbf{H}_P^T}{\sqrt{D}}\right)\mathbf{H}_P \quad (5) \\ &= \mathbf{A}\mathbf{H}_P \in \mathbb{R}^{F \times D}, \end{aligned}$$

where $\mathbf{A} \in \mathbb{R}^{F \times L}$ represents the attention weight matrix, $L$ denotes the length of the given phoneme sequence. Following the attention module, the expanded phoneme hidden representation is obtained by linear combination. A residual connection is employed to integrate $\mathbf{H}_A$ for efficient training, with a dropout layer to prevent acoustic features from excessively relying on articulation information during training .

*3.2.3 Training Criterion.* The proposed pseudo target generation module is trained on vocalized utterances because they simultaneously contain text, articulatory movements, and corresponding speech. Since the spectrogram denoiser generates synchronized acoustic features conditioned on $\mathbf{C}$, the pseudo target generation module employs the same loss function as Eq. 4.

To assist in learning the alignment between text information and articulatory movements, we additionally propose supervising the model using an attention loss instead of the diagonal loss in most automatic video dubbing studies. Specifically, we employ the following L1 loss function

$$\mathcal{L}_A = ||\mathbf{A} - \hat{\mathbf{A}}||_1 \quad (6)$$

where $\hat{\mathbf{A}}$ represents the generated attention matrix by the text-articulation aligner, and $A$ denotes its groudtruth value. To acquire the groundtruth value $A$, we utilize the montreal forced aligner (MFA) tool[2] [31] to obtain the alignment between phoneme sequences and real speech frames. Considering the correspondence between speech and video frame rates (set to be the same in our experiments), we further obtain the alignment between phoneme sequences and articulation frames, serving as the ground truth value for the attention matrix $\mathbf{A}$.

Therefore, the overall loss function for training the pseudo target generation module can be expressed as:

$$\mathcal{L} = \mathcal{L}_\theta + \mathcal{L}_A. \tag{7}$$

Once the pseudo target generation module is trained on vocalized utterances, we apply it to silent articulation and generate pseudo acoustic features for training A2A conversion model in the silent mode accordingly.

## 3.3 Model Training

Our A2A conversion model training approach includes two steps. Firstly, considering the absence of speech and the limited articulation data in silent mode, the proposed A2A conversion model and the dubbing-based pseudo target generation module are initially trained on vocalized utterances, where corresponding speech can be used as targets. Secondly, as the model trained on vocalized utterances cannot be directly applied to silent articulation due to domain discrepancy, we further train the proposed A2A conversion model on silent articulation with pseudo acoustic features generated by the trained dubbing module. We also propose a combined training approach to enhance the model's performance on silent articulation. The detailed training approach is described below.

### 3.3.1 First Step: Training on Vocalized Articulation.
While training the proposed A2A conversion model on vocalized utterances, we employ a transfer learning strategy which has proved to be effective in TaLNet [49]. Specifically, a multi-speaker TTS model is initially trained on a large multi-speaker TTS corpus. This multi-speaker TTS model shares a similar architecture with the proposed A2A conversion model, except that the articulation encoder is replaced by a text encoder and a variance adaptor in FastSpeech2 [35]. After obtaining the pre-trained TTS model, its spectrogram denoiser is transferred as that of the A2A conversion model, which is then jointly trained with the articulation encoder on the TaL corpus [37].

The proposed dubbing-based pseudo target generation module is also trained on vocalized utterances. Before training, we initialize its articulation encoder and the rest parts with the pre-trained vocalized A2A conversion model and TTS model, respectively. The initialization makes it easier for the dubbing module to align text with articulatory movements compared to learning from scratch.

### 3.3.2 Second Step: Training on Silent Articulation.
After learning from vocalized utterances, the proposed model is further trained on silent articulation. Before training, we initialize the silent A2A conversion model with the pre-trained vocalized model. Pseudo acoustic features, generated by the trained dubbing module based on

[2]https://github.com/MontrealCorpusTools/Montreal-Forced-Aligner

the provided text, lip videos, and ultrasound tongue images recorded in the silent speaking mode, are used as supervision targets.

Notably, after acquiring the pseudo acoustic features, a filtering process is conducted. An automatic speech recognition (ASR) engine is employed to transcribe the pseudo speech transformed from the generated acoustic features using a vocoder. Utterances with a word error rate (WER) surpassing a predefined threshold are omitted from the training set. This exclusion is justified by the presumption that such cases potentially indicate articulation errors deriving from the absence of auditory feedback in silent mode.

A combined training approach is also adopted to train the proposed A2A conversion model on silent articulation. This approach involves combining pseudo acoustic features generated by both the proposed dubbing strategy and the previous DTW strategy as supervision targets. Specifically, during training, the pseudo acoustic features generated by the proposed dubbing strategy are selected as the supervisory target with a probability of $p$, while the pseudo acoustic features generated by the DTW strategy proposed in the previous study [50] are chosen with a probability of $1 - p$.

## 4 EXPERIMENTS

### 4.1 Datasets

The TaL80 subset of the TaL dataset [37] was utilized in our experiments, comprising 14,257 utterances in the vocalized speaking modes from 81 native English speakers. Each utterance includes corresponding text, synchronized audio, ultrasound tongue images, and lip videos. Additionally, it contains 1,212 utterances in the silent speaking mode, each accompanied by corresponding text, ultrasound tongue images, and lip videos. We adopted the same training, validation, and testing set partitioning described in [49] for vocalized utterances and [50] for silent utterances, respectively.

### 4.2 Implementation Details

Consistent with previous studies [49, 50], we utilized mel-spectrograms as acoustic features and followed the data processing pipeline outlined in [49] to obtain the lip videos, ultrasound tongue images, and mel-spectrograms as model inputs. We employed a well-trained Parallel WaveGAN (PWG) [46] vocoder to transform the synthesized mel-spectrograms into speech waveforms for fair comparison with [49, 50]. Considering the limited number of silent utterances for each speaker, we developed our proposed model in a speaker-independent manner without further fine-tuning using speaker-dependent data. We extracted speaker embedding using the DeepSpeaker system [23] for speaker representation.

We used the discretization of the continuous-time extension of the diffusion process in Eq. 3 with the variance preserving (VP) SDE [40] to compute the noise schedule $\{\beta_t\}_{t=1}^{T}$ for the spectrogram denoiser in both the proposed A2A conversion model and the pseudo target generation module. Specifically, we set step $T = 4$, and computed $\{\beta_t\}_{t=1}^{4}$ as:

$$\beta_t = 1 - \exp\left(-\frac{\beta_{min}}{T} - 0.5(\beta_{max} - \beta_{min})\frac{2t - 1}{T^2}\right), \tag{8}$$

where $\beta_{min}$ and $\beta_{max}$ were set to be 0.1 and 40 respectively.

The diffusion-based multi-speaker TTS model described in Section 3.3.1 was trained on 460 hours data from 1150 speakers of

**Table 1: Objective and subjective evaluation results of speech reconstructed in vocalized and silent speaking modes. *Vocoder* represents the vocoder-resynthesized natural speech in the vocalized speaking mode. Best results are highlighted in bold. All results are the means on the test set. ± represents 95% confidence intervals.**

| Method | Mode | MCD/dB | F0 RMSE/Hz | STOI | ESTOI | WER/% | CER/% | MOSNet | MOS |
|---|---|---|---|---|---|---|---|---|---|
| GroundTruth | | / | / | / | / | 3.80 | 2.13 | 4.31 | 4.02±0.06 |
| Vocoder | Vocalized | 1.92 | 13.49 | 0.94 | 0.88 | 5.29 | 3.07 | 4.31 | 3.93±0.06 |
| TaLNet [49] | | 3.37 | 25.13 | 0.69 | 0.52 | **36.93** | 25.40 | 3.86 | 3.37±0.07 |
| Proposed | | **3.22** | **24.98** | 0.69 | 0.52 | 37.18 | **25.36** | **4.25** | **3.46±0.07** |
| TaLNet [49] | | 4.15 | 32.66 | 0.32 | 0.14 | 87.24 | 69.79 | 3.81 | 3.27±0.08 |
| Zheng et al. [50] | Silent | 3.78 | 33.74 | 0.31 | 0.15 | 78.24 | 58.31 | 3.71 | 3.35±0.08 |
| Proposed | | **3.53** | **31.06** | **0.37** | **0.22** | **73.32** | **54.67** | **3.99** | **3.49±0.08** |

the LibriTTS corpus [48] using an Adam optimizer with an initial learning rate $1e-4$ for 300k steps. The proposed A2A conversion model and the pseudo target generation module were first trained on vocalized utterances with an Adam optimizer whose learning rate was dynamically adjusted as

$$lr = D^{-0.5} * min(step^{-0.5}, step \times warmup^{-1.5}), \quad (9)$$

where $D = 512$ denotes the feature dimension of the hidden representation, $step$ represents the training step, and $warmup = 30,000$ respresents warmup steps. Furthermore, after transferring the spectrogram denoiser from the TTS model to the A2A conversion model, its parameters were frozen for 30k steps and then optimized together with the other parts in subsequent steps. For training the A2A conversion model on silent utterances, the threshold for filtering utterances with articulation errors and the probability $p$ in the combined training strategy described in Section 3.3.2 was empirically set to 40% and 0.5, respectively. An Adam optimizer with an initial learning rate of $1e-4$ and a learning rate exponential decay strategy was adopted. Specifically, the learning rate decayed by a factor of 0.999 at the end of each epoch. The batch size for training the A2A conversion model was 16, while the batch size for training the pseudo target generation module was 8. All experiments were conducted on an NVIDIA GeForce GTX 3090 GPU.

## 4.3 Evaluation Metrics

We included TaLNet [49] and the method proposed by Zheng et al. [50], which were previous state-of-the-art methods on the TaL dataset [37] in vocalized and silent modes, as baselines for comparison. The effectiveness of our proposed method were assessed through both objective and subjective evaluations.

*4.3.1 Objective Evaluation.* For objective evaluation, mel-cepstral distortion (MCD), F0 root mean squared error (F0 RMSE), short-term objective intelligibility (STOI), and extended STOI (ESTOI) were used as metrics. Since ground truth speech for the silent utterances was unavailable, we used the speech corresponding to the vocalized utterance from the same speaker with consistent linguistic content as the reference speech. Before evaluation, we aligned the generated speech with the reference speech using DTW. In addition to these metrics, WER and character error rate (CER) from an ASR engine were computed. We utilized the ASR API provided in ESPNet[3] [45] to transcribe the synthesized speech. Furthermore,

to evaluate the naturalness of the synthesized speech, we employed an automatic speech quality assessment system, MOSNet [27], to assign naturalness scores.

*4.3.2 Subjective Evaluation.* Two groups of subjective listening tests were also conducted to measure the naturalness mean opinion scores (MOS) of reconstructed speech in the two modes, respectively. In each test, thirty native English speakers were recruited on Amazon's Mechanical Turk[4] and were asked to give a 5-point score (1-very poor, 2-poor, 3-fair, 4-good, 5-excellent) for each utterance they listened to. Twenty utterances in the vocalized mode and fifteen in the silent mode generated by each system were randomly selected for MOS evaluation.

## 4.4 Experimental Results

*4.4.1 Main Results.* We first present the results of the proposed diffusion-based A2A conversion model on speech reconstruction from ultrasound tongue images and lip videos in the vocalized speaking mode, as shown in the top four rows in Table 1. We can see that the proposed A2A conversion model significantly improved the naturalness of the speech reconstructed from vocalized articulation compared to TaLNet [49]. Specifically, when using MOSNet to assess the naturalness of the generated speech, the score increased by approximately 10%. An increase of subjective MOS by approximately 0.1 ($p = 1.39 \times 10^{-2}$ in paired t-test) is also observed.

The evaluation results in the silent speaking mode are exhibited in the last three rows of Table 1. These results show that our proposed method outperformed all baselines across all metrics, demonstrating its effectiveness. Specifically, when comparing the proposed method with the previous state-of-the-art method in the silent speaking mode by Zheng et al. [50], a notable increase of subjective MOS by 0.14 ($p = 1.12 \times 10^{-2}$ in paired t-test) is observed, along with a further 5% decrease in WER, indicating superior intelligibility and naturalness of the reconstructed speech.

We also present spectrogram visualizations of the speech generated by various systems when provided with identical lip and tongue articulation inputs in both vocalized and silent speaking modes, as depicted in Fig. 3. In comparison to non-probabilistic models like TaLNet [49], our proposed diffusion-based A2A conversion model tends to produce speech with less over-smoothing spectrograms for vocalized utterances, thus yielding more natural speech. Moreover, our method for silent articulation demonstrates

---

[3]https://github.com/espnet/espnet_model_zoo

[4]https://www.mturk.com/

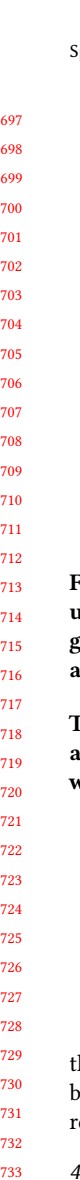
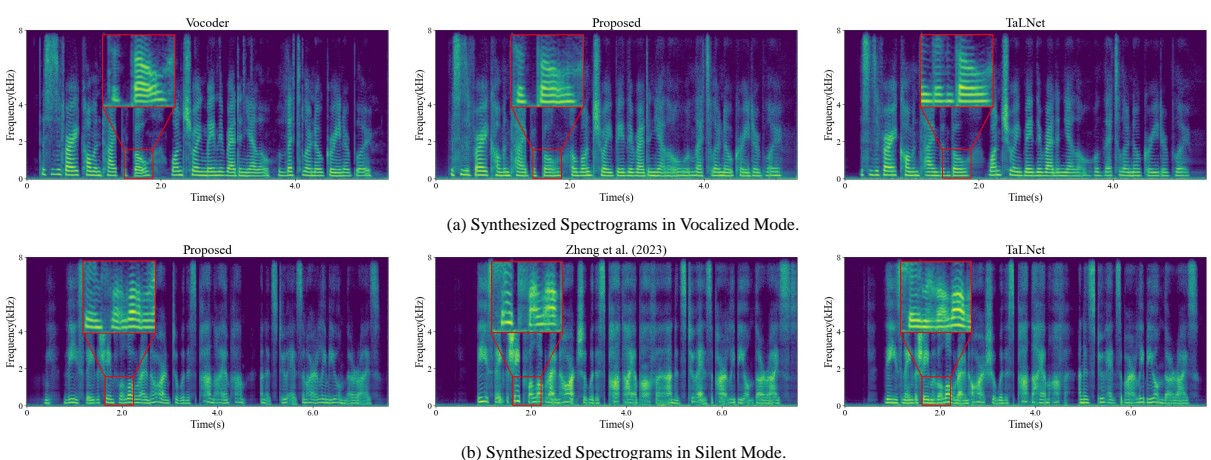

(a) Synthesized Spectrograms in Vocalized Mode.

(b) Synthesized Spectrograms in Silent Mode.

**Figure 3: Visualizations of the generated spectrograms by different systems in both vocalized and silent modes. For the vocalized utterance, the corresponding text is "This is a very common type of bow, one showing mainly red and yellow, with little or no green or blue". For the silent utterance, the corresponding text is "These take the shape of a long round arch, with its path high above, and its two ends apparently beyond the horizon".**

**Table 2: Objective evaluation results of the proposed method on the task of speech reconstruction from silent lip and tongue articulation in ablation studies. "w/o pseudo targets" represents the proposed model trained solely on vocalized utterances without further training with pseudo targets on silent articulation. Best results are highlighted in bold.**

| Method | MCD/dB | F0 RMSE/Hz | STOI | ESTOI | WER/% | CER/% | MOSNet |
|---|---|---|---|---|---|---|---|
| Proposed | 3.53 | **31.06** | **0.37** | **0.22** | **73.32** | **54.67** | **3.99** |
| w/o Pseudo Targets by DTW | 3.64 | 34.79 | **0.37** | 0.20 | 79.60 | 61.74 | 3.88 |
| w/o Pseudo Targets by Dubbing | **3.52** | 31.29 | 0.36 | 0.20 | 76.18 | 58.37 | 3.97 |
| w/o Filtering Pseudo Targets | 3.70 | 37.08 | 0.35 | 0.18 | 79.64 | 62.43 | 3.86 |
| w/o Pseudo Targets | 3.82 | 36.40 | 0.36 | 0.19 | 88.48 | 70.32 | **3.99** |

the capability to generate speech with more reasonable phoneme boundaries while maintaining a diverse set of samples, ultimately resulting in improved intelligibility of the reconstructed speech.

*4.4.2 Ablation Studies.* Ablation studies were conducted to examine the effectiveness of each part in our proposed training approach for silent speaking mode. All the results are presented in Table 2.

Specifically, we evaluated the contribution of the combined training approach by comparing the performance of the proposed model trained with mel-spectrograms generated by different pseudo target generation strategies, as demonstrated in the top three rows. It is evident that the model trained solely with the pseudo targets generated by either the dubbing strategy or the DTW strategy fails to achieve optimal performance. Moreover, we include the results of the proposed model trained on vocalized utterances without further training with pseudo targets on silent articulation in the last row. A comparison between the second row and the last row reveals that training the proposed model with pseudo mel-spectrograms generated by the dubbing strategy notably enhances the model's performance on silent articulation, demonstrating the efficacy of the proposed dubbing strategy. Furthermore, we analyzed the benefits of the filtering process proposed in Section 3.3.2, and the results of training the proposed model with unfiltered pseudo targets are exhibited in the fourth row. It reflects that not filtering the generated pseudo targets yields inferior performance. Additionally, comparing

the last row of Table 2 with the fifth row of Table 1, we observe that the proposed diffusion-based A2A architecture outperforms TaL-Net [49] on most evaluation metrics, even without further pseudo targets training on silent articulation. This further showcases the superior generative capabilities of the proposed diffusion-based framework over non-probabilistic models in silent speaking mode.

*4.4.3 Analysis of the Pseudo Targets Generated by Different Strategies.* To delve deeper into the effectiveness of the proposed dubbing strategy, we conducted supplementary experiments. We argue that the effectiveness of the proposed dubbing strategy lies in its ability to generate pseudo acoustic features well-synchronized with the provided silent articulatory movements. To validate this argument, we evaluated the pseudo speech (transformed from the acoustic features using a PWG vocoder) generated by different pseudo target generation strategies from two distinct perspectives: speech naturalness and synchronicity between speech and articulatory movements. We employed MOSNet scores to assess the naturalness of the generated speech. Additionally, we utilized a well-trained SyncNet [2] model to measure the lip-sync error between the generated speech and the provided silent lip videos, following the evaluation metrics proposed by Prajwal et al. [34]. The first evaluation metric is Lip Sync Error - Distance (LSE-D), representing the average error measure calculated in terms of the distance between the lip and audio representations. A lower LSE-D denotes a

**Table 3: Analysis results of the pseudo targets generated by different strategies. Mode indicates the speaking mode in which pseudo target generation strategy is utilized to generate synthesized speech (transformed from acoustic features with PWG vocoder).**

| Method | Mode | MOSNet | LSE-D($\downarrow$) | LSE-C($\uparrow$) |
|---|---|---|---|---|
| DTW | Silent | 4.05 | 9.79 | 1.42 |
| Dubbing | | 3.99 | 9.67 | 1.50 |
| Dubbing | Vocalized | 4.19 | 9.122 | 1.96 |

higher audio-visual match. The second metric is Lip Sync Error - Confidence (LSE-C), denoting the average confidence score. The higher the confidence, the better the audio-video correlation. Since the SyncNet model available online is pre-trained on entire face images, but the TaL80 dataset only contains lip videos of speakers, we trained the SyncNet model on the TaL80 dataset following the instructions provided in [32][5]. The results are shown in Table 3. We have also included evaluation results of speech generated by the dubbing strategy on vocalized articulation to demonstrate the performance of the proposed dubbing module when both training and testing data originate from the same vocalized domain.

The results indicate that the dubbing strategy outperforms the DTW strategy in lip-sync synchronicity. In contrast, while the speech generated by the DTW strategy exhibits commendable naturalness, its synchronicity falls short compared to the proposed dubbing strategy. This difference arises from the fundamental approach of the dubbing strategy, which establishes a correspondence between text and silent articulation. By identifying the boundaries of articulatory movements corresponding to the text, it generates speech from the text that is highly synchronized with these movements. Conversely, the DTW strategy, relying on DTW alignment between vocalized and silent articulation representations, may encounter alignment failures due to the discrepancy between vocalized and silent articulation. As a result, though the pseudo mel-spectrograms derived by aligning vocalized mel-spectrograms from the DTW strategy ensure no missing and repeated linguistic content, even in cases of alignment failure, they do not guarantee precise synchronization between pseudo speech and articulatory movements. Therefore, a combination of targets generated by the DTW strategy, which prioritizes speech naturalness, with those from the dubbing strategy, which emphasizes articulation synchronization, could enhance speech reconstruction performance in the silent speaking mode. Moreover, the second and third rows of Table 3 show a performance decline when the pseudo target generation module trained on vocalized utterances is directly applied to silent articulation, which can be attributed to two reasons. One is due to the difference between the training and test data. The other is that the lack of audio feedback for speakers in the silent speaking mode potentially leads to articulation errors and alignment failure. Consequently, generating appropriate speech for these silent utterances based on the provided text and the articulatory movements becomes challenging. In our proposed method, we use a filtering process described in 3.3.2 to exclude the impact of these utterances on model performance during training.

[5]https://github.com/joonson/syncnet_trainer

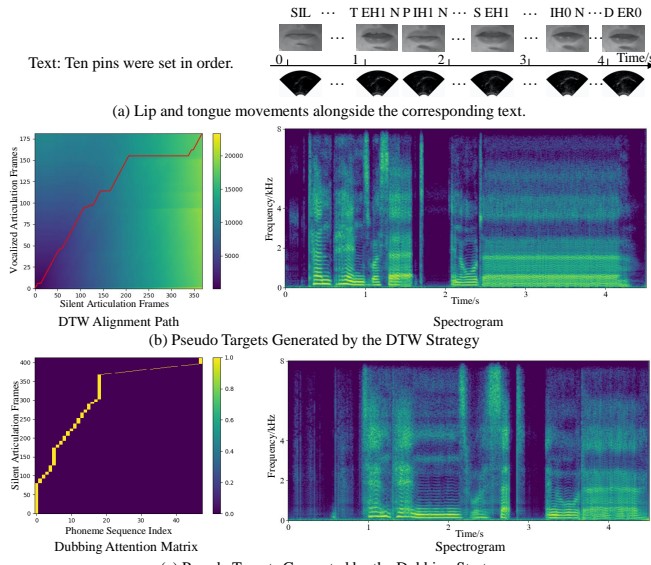

(a) Lip and tongue movements alongside the corresponding text.

(b) Pseudo Targets Generated by the DTW Strategy

(c) Pseudo Targets Generated by the Dubbing Strategy

**Figure 4: An example of pseudo mel-spectrograms generated for silent utterances using both the DTW and Dubbing pseudo target generation strategy. The correspondence between the articulatory movements and the phoneme sequence in (a) is manually annotated. The frame rate of the lip video and ultrasound tongue images is 81.5fps.**

Fig. 4 illustrates an example of generating pseudo mel-spectrograms for silent utterances using DTW and the proposed dubbing strategies respectively. Specifically, the speaker starts speaking around 1 second, as evidenced by the lip and tongue movements from Fig. 4(a). However, in Fig. 4(b), though the DTW strategy generates the mel-spectrogram with correct linguistic content, its synchronization with the corresponding silent lip and tongue movements is deficient due to the errors in the obtained DTW alignment path, causing the generated mel-spectrogram to start displaying linguistic content from around 0.3 seconds. Conversely, leveraging text information, the dubbing strategy yields more reliable alignment paths, particularly during silence segments, resulting in pseudo mel-spectrograms with significantly enhanced synchronization as shown in Fig. 4(c). More examples are available at our project page.

## 5 CONCLUSION

This paper solves the task of speech reconstruction from ultrasound tongue images and lip videos in the silent speaking mode. We propose a diffusion-based A2A conversion model and introduce a novel text-guided pseudo target generation strategy, producing pseudo acoustic features for the supervised training of the proposed model on silent articulation. Experimental results demonstrate the effectiveness of the proposed method in enhancing the naturalness and intelligibility of the speech reconstructed from the silent lip and tongue articulation. Our future endeavors will focus on developing real-time systems for speech reconstruction from silent articulation, aiming to provide speakers with auditory feedback during silent articulation processes.

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
