# OpenReview forum: "Speech Reconstruction from Silent Lip and Tongue Articulation by Diffusion Models and Text-Guided Pseudo Target Generation"
_acmmm.org/ACMMM/2024/Conference — MM2024 Poster_

### Official Review · Reviewer_Q9Xk · 2024-05-24

**Rating:** 4
**Confidence:** 3

**Summary:**

This paper studies the task of speech reconstruction from ultrasound tongue images and optical lip videos recorded in a silent speaking mode. To overcome the domain discrepancy between silent and standard vocalized articulation, this paper introduces a pseudo target generation strategy. This strategy integrates the text modality to align with articulatory movements, thereby guiding the generation of pseudo acoustic features for supervised training on speech reconstruction from silent articulation. Experimental results show that the proposed method significantly improves the intelligibility and naturalness of reconstructed speech in the silent mode. The proposed pseudo target generation strategy can generate pseudo acoustic features that better synchronize with articulatory movements.

**Strengths:**

1. The authors integrate the text modality to guide the generation of pseudo acoustic features for supervised training on speech reconstruction from silent articulation. By learning the alignment between text and articulation, the dubbing strategy generates pseudo acoustic features synchronized better with the given articulatory movement than previous method while maintaining content consistency with the provided text, thereby improving the overall task performance.
2. They propose a combined training approach to further improve the naturalness and intelligibility of the reconstructed speech in the silent mode.
3. The writing in this paper is fluent, and the vocabulary is appropriately used.

**Limitations:**

1. This paper propose supervising the model using an attention loss instead of the diagonal loss in most automatic video dubbing studies. However, no comparative experiments were conducted on the training effects of the two loss functions.
2. The 4th formula should be changed to $\mathcal{L}_{\theta}=\|f_{\theta}(\alpha_tx_0+\sqrt{1-\alpha_t^2}\epsilon|t,s,H_A)-x_0\|_2^2,\epsilon\sim N(0,I)$.
3. There are some formatting errors in the paper, such as the overflow of $p_{\theta}(x_{t-1}|x_t)$ beyond the document boundaries in the second paragraph of section 3.1.2.

**Suitability:**

2

---

### Official Review · Reviewer_9At9 · 2024-05-24

**Rating:** 6
**Confidence:** 3

**Summary:**

This paper explores the task of converting ultrasound tongue images and optical lip videos, recorded during silent speech, into audible speech. Due to the absence of ground-truth target speech, the authors propose a pseudo target generation strategy. This strategy produces pseudo speech labels for silent articulation. Additionally, a diffusion model is employed to map the ultrasound tongue images and optical lip videos into speech acoustic features. A transfer learning training strategy is also utilized. Experimental results reveal that the proposed method outperforms state-of-the-art methods in both objective and subjective evaluations.

**Strengths:**

- This paper presents a unique method for reconstructing speech from ultrasound tongue images and optical lip videos recorded during silent speech. This approach utilizes state-of-the-art generative models (DDPMs) to produce speech from silent articulation and proposes a new method for acquiring pseudo speech labels for silent articulation.
- The experiments highlight the superiority of this model over other state-of-the-art methods through both subjective and objective evaluations. A thorough ablation study validates the effectiveness of these designs. The analysis of the use of Dynamic Time Warping (DTW) and the proposed dubbing strategy for generating pseudo speech labels provides valuable insights.
- The paper is well-structured and easy to understand.

**Limitations:**

- An attention matrix supervision from force-alignment is introduced during the dubbing model's training. However, an ablation study regarding this design is lacking.
- Additionally, regarding the alignment obtained from the force-alignment, why not use the alignment labels to train a dubbing model? This could predict the duration of each text token and generally provide a more reliable alignment than the attention mechanism.

**Suitability:**

3

---

### Official Review · Reviewer_21RP · 2024-05-25

**Rating:** 3
**Confidence:** 3

**Summary:**

This paper focuses on speech reconstruction given silent lip and tongue articulation. Under this setting, data samples are collected when people move their lips to say words but without producing sound, unlike traditional lip-to-speech methods that rely on parallel lip and speech data. The paper proposes a new method to generate pseudo speech ground truth for silent lips. Compared to an existing method for pseudo speech generation, DTW, this method utilizes text modality to improve the quality of the pseudo speech. Additionally, for speech reconstruction, the authors use a diffusion-based method to enhance speech synthesis.

**Strengths:**

1. The idea of estimating pseudo speech ground truth using extra text modality is interesting.
2. The ablation study is comprehensive.
3. The paper is well-organized.
4. The diffusion model does improve speech synthesis.

**Limitations:**

From the results, it appears that the proposed method of pseudo speech generation, dubbing, shows only marginal improvement over DTW, but the proposed method requires significantly more effort to implement. In Table 2, the methods without dubbing maintain almost constant performance compared to those with dubbing, except for content metrics like WER and CER. However, if DTW is removed, the model's performance significantly worsens. Additionally, Table 3 compares DTW and dubbing on audio-visual synchronization, and based on my knowledge, the difference in LSE is not significant. If dubbing is not strong enough, its contribution is inefficient, given that it is the key motivation of the paper.

**Suitability:**

3

---

### Official Review · Reviewer_SrfG · 2024-06-01

**Rating:** 4
**Confidence:** 3

**Summary:**

The paper presents a novel method for speech reconstruction from silent lip and tongue articulation using a text-guided pseudo target generation strategy and a denoising diffusion probabilistic model. Unlike previous approaches, it integrates text to better align with articulatory movements, improving synchronization and performance. The method employs a combined training approach, leveraging both new and traditional pseudo target generation strategies. This results in significant improvements in the naturalness and intelligibility of reconstructed speech, simplifying and enhancing the overall process.

**Strengths:**

The proposed method excels in several areas: it introduces a novel text-guided pseudo target generation strategy, improving synchronization between speech and articulatory movements. Utilizing a denoising diffusion probabilistic model, it enhances the naturalness and intelligibility of reconstructed speech. The combined training approach leverages strengths from multiple strategies, yielding better performance. Additionally, the method simplifies the pseudo target generation process by avoiding complex adversarial training, making it more efficient and practical for real-world applications.

**Limitations:**

The proposed method has some limitations: it is complex and computationally intensive, posing challenges for practical implementation and scalability. It relies heavily on high-quality articulatory data, which can be difficult to obtain consistently in real-world scenarios. The evaluation is limited to a specific dataset, raising questions about generalizability. Additionally, the filtering process may exclude valuable data, potentially affecting robustness. The lack of real-time performance evaluation and handling of articulation errors in silent mode are also significant limitations.

**Suitability:**

3

---

### Meta-Review · Area_Chair_jdaJ · 2024-07-03

**Recommendation:** Accept (Poster)
**Confidence:** 4

**Metareview:**

All reviewers agree that the proposed method is interesting and innovative. All reviewers comment that integrating the text modality to guide the generation of pseudo acoustic features for supervised training on speech reconstruction from silent articulation is novel. The reviewers are satisfied with the presented experimental study. The rebuttal addressed a number of additionally raised questions. The authors are advised to include the relevant explanation in the final version of the paper. Given a general appreciation of the work by the reviewers, I believe that the paper will be of interest to the audience attending ACM MM and would recommend a presentation of the work as a poster.